# The Deubiquitinase USP39 Promotes Esophageal Squamous Cell Carcinoma Malignancy as a Splicing Factor

**DOI:** 10.3390/genes13050819

**Published:** 2022-05-03

**Authors:** Xiaolin Zhu, Jianlin Ma, Minyi Lu, Zhihua Liu, Yongkun Sun, Hongyan Chen

**Affiliations:** 1State Key Laboratory of Molecular Oncology, National Cancer Center/National Clinical Research Center for Cancer/Cancer Hospital, Chinese Academy of Medical Sciences and Peking Union Medical College, Beijing 100021, China; zhuxiaolinyy@163.com (X.Z.); jianlin1234@126.com (J.M.); luminyi_2629@163.com (M.L.); liuzh@cicams.ac.cn (Z.L.); 2Department of Medical Oncology, National Cancer Center/National Clinical Research Center for Cancer/ Cancer Hospital, Chinese Academy of Medical Sciences and Peking Union Medical College, Beijing 100021, China; 3Department of Medical Oncology, National Cancer Center/National Clinical Research Center for Cancer/Hebei Cancer Hospital, Chinese Academy of Medical Sciences, Langfang 065001, China

**Keywords:** ESCC, USP39, splicing factor

## Abstract

Esophageal squamous cell carcinoma (ESCC) is an aggressive epithelial malignancy and the underlying molecular mechanisms remain elusive. Here, we identify that the ubiquitin-specific protease 39 (USP39) drives cell growth and chemoresistance by functional screening in ESCC, and that high expression of USP39 correlates with shorter overall survival and progression-free survival. Mechanistically, we provide evidence for the role of USP39 in alternative splicing regulation. USP39 interacts with several spliceosome components. Integrated analysis of RNA-seq and RIP-seq reveals that USP39 regulates the alternative splicing events. Taken together, our results indicate that USP39 functions as an oncogenic splicing factor and acts as a potential therapeutic target for ESCC.

## 1. Introduction

Esophageal cancer has become the sixth most common cancer diagnosed in China and the fourth leading causes of cancer death [1]. The overall 5-year survival of patients ranges from 15 to 25% [2]. Esophageal squamous cell carcinoma (ESCC) is the predominant histological subtype of esophageal cancer. Moreover, China alone contributed more than half of the global esophageal squamous cell carcinoma cases [3]. Patients diagnosed with ESCC have reported poor prognoses. Chemoresistance is a major problem in the treatment of ESCC patients and leads to poor prognosis. Our recent studies showed that decreased ARID1A or OTUD1 promote chemoresistance and proposed the synthetically lethal strategy in ESCC [4,5]. However, the molecules that are highly expressed and positively correlated with chemoresistance may have a potential to be targeted and require further investigation.

USP39 is a member of the deubiquitylation family. Emerging evidence revealed that the aberrant expression of USP39 plays vital roles in tumorigenesis and tumor progression in various cancer types [6,7,8,9]. Intriguingly, USP39 is considered to be devoid of deubiquitinating activity due to the absence of active site residues [10]. Nevertheless, it has been demonstrated that USP39 can regulate ubiquitination of ZEB1, CHK2 and SP1 [6,9,11]. Currently, USP39 is well known as an essential factor for the RNA splicing by recruitment of U4/U6-U5 tri-snRNP [12]. As a splicing factor, USP39 was reported to be a regulator of *EGFR* pre-mRNA splicing [13]. USP39 has also been implicated in the mRNA maturation of *TAZ* and *VEGF-A165b* [14,15].

Almost all human mRNAs are required to be spliced for maturation. RNA splicing is critical for processing pre-mRNA transcribed from protein-coding genes that contain more than one exon into mature mRNAs prior to translation into proteins [16,17]. A previous study indicates that alternative splicing occurs with genes involved in almost every aspect of cancer cell biology, such as proliferation, metastasis, and apoptosis [18]. Alternative splicing events were determined to be significantly associated with the overall survival of patients in ESCC [19]. Splicing factors such as SF3B4 were found to play a crucial role in the lymphatic progression of ESCC [20]. Furthermore, a previous study identified 14710 aberrant alternative splicing events in ESCC, 92.67% of them affecting coding genes. Interestingly, the regulatory network of RNA-binding protein and aberrant alternative splicing events were enriched in apoptotic pathways and various adhesion–junction-related processes in ESCC [21].

Here, we found that USP39 promotes cancer cell growth and chemoresistance in ESCC. We next identified that USP39 interacted with several spliceosome components and regulated alternative splicing events. Our findings highlight the biological significance of USP39 in ESCC and provide a promising therapeutic target against human ESCC.

## 2. Material and methods

### 2.1. Cell Culture and Plasmids

Human esophageal squamous cell carcinoma (ESCC) of KYSE series cell lines (KYSE30, KYSE410, and KYSE 450) were provided by Dr. Yutaka Shimada (Kyoto University, Japan). 293T cell lines were purchased from the American Type Culture Collection (ATCC, Manassas, VA, USA). KYSE30, KYSE410, and KYSE450 were cultured in RPMI-1640 medium supplemented with 10% fetal bovine serum (FBS). 293T cells were cultured in DMEM supplemented with 10% FBS.

We designed the PCR primers of USP39 CDS and cloned the full length of USP39. Then, DNA was digested by restriction endonucleases. Next, the fragment and vector were ligated by T4 DNA ligase. The USP39 CDS was cloned into pLVX-IRES-Neo (632181, Clontech, Mountain View, CA, USA) vector.

The shRNA sequences targeting USP39 and PSMD14 were synthesized from the Beijing Genomics Institution (BGI). After annealing assays, the shRNA sequences were cloned into the vector pSIH1-H1-Pure (#26597, Addgene, Watertown, MA, USA) vector. Sequences and shRNAs were listed in Appendix A.

### 2.2. Antibodies and Reagents

The antibodies used for immunoblotting, immunoprecipitation, immunochemistry, and immunofluorescence in this study were as follows: anti-USP39, 1:1000 (IB; Abcam, Waltham, MA, USA, ab131244); 1:100 (IHC; Abcam, Waltham, MA, USA, ab131244); (RIP, Proteintech, Rosemont, IL, USA, 23865-1-AP); anti-Flag-Tag, 1:1000 (IB; #80010-1-RR, Proteintech, Rosemont, IL, USA); anti-His-Tag, 1:1000 (IB; Cell Signaling Technology, #12698); anti-PARP, 1:1000 (IB; Cell Signaling Technology, Danvers, MA, USA, #9542); anti-Caspase-3, 1:1000 (IB; Cell Signaling Technology, Danvers, MA, USA, #9662); and anti-β-actin, 1:5000 (IB: Sigma, Louis, MO, USA, A5316).

### 2.3. Virus Production and Cell Infection

Lentivirus was produced by co-transfecting 293T cells with the generation packaging system, including psPAX2 (#12260, Addgene) and pMD2.G (#12259, Addgene). The virus-containing medium was collected 48 h after transfection. Before virus infection, the cell density should be about 30–40%. Then, the medium was added to the cells supplemented with 8 μg/mL of polybrene. After 48 h, infected cells were selected with 1 μg/mL of puromycin or 200 μg/mL of G418 for a week.

### 2.4. Screening of DUB Library

The siGENOME human DUB enzyme including 98 deubiquitinase was purchased from Thermo Fisher (Carlsbad, CA, USA), and the siRNA pool consisted of four siRNA sequences.

### 2.5. Human Esophageal Tumor Tissue Collection and Immunohistochemistry Staining

In total, 199 ESCC samples were obtained from Zhejiang Cancer Hospital (Hangzhou, China). The study was approved by the ethical committee of the Cancer Hospital, Chinese Academy of Medical Sciences and Zhejiang Cancer Hospital, and all patients provided informed consent. The information of clinical samples is shown in Appendix A. The samples were used to detect USP39 expression by immunohistochemistry (IHC). Human esophageal cancer tissues were fixed in 4% (*v*/*v*) formaldehyde in PBS and then embedded in paraffin and cut into 5 μm sections. Tissue sections were dewaxed in xylene and dehydrated in an alcohol gradient of 95%, 85%, and 70%. Antigen retrieval was achieved by heating slides covered with citrate buffer at 95 °C for 10 min. Then, 10% goat serum albumin was used to block nonspecific binding, followed by incubation with primary antibodies at 4 °C overnight in a moist chamber. After washing three times with phosphate-buffered saline (PBS), the slides were incubated with a secondary antibody at room temperature for 1 h and washed with PBS. Diaminobenzidine (DAB) was used as a chromogen, and the sections were counterstained with hematoxylin. All slides were scanned using an Aperio scanning system (Aperio, San Diego, CA, USA). IHC results were scored with H-Score [22]; a final score was obtained by multiplying the scores of “percentage of USP39-positive cells” and staining intensity (assigned on 4 levels: 0, negative; 1, weak staining; 2, mild staining; 3, strong staining). The staining intensity was evaluated by two pathologists, independently.

### 2.6. CCK-8 cell Proliferation Assays

CCK-8 reagent (Dojindo, Tokyo, Japan) was added to the cell culture medium at a ratio of 1:10 pre-use, and co-incubated for 1 h at 37 °C. Absorbance values were measured at 450 nm using a microplate reader (BioTek, Winooski, VT, USA). The relative cell proliferation rate was calculated as follows: the average absorbance at the control group was designated as 1. The relative cell proliferation rate is the ratio of the absorbance of each experimental group to the average of the control group.

### 2.7. Flow Cytometry

Flow cytometry was performed using an apoptosis detection kit (Dojindo, Tokyo, Japan) according to the manufacturer’s protocol.

### 2.8. Cloning formation assay

A total of 3000 cells were suspended in culture medium and seeded in 6-well plates. The cells were cultured at 37 °C until the colony formation. Then, the cells were stained by crystal violet solution and the number of colonies was counted.

### 2.9. Xenograft Transplantation Experiments

The Animal Care and Use Committee of the Chinese Academy of Medical Sciences Cancer Hospital approved all animal protocols. For the tumor formation assay, 8 × 10^6^ KYSE30 cells were injected subcutaneously into the right armpit of 6-week-old male BALB/c nude mice. The tumor size was calculated as 0.5 × length × width^2^.

### 2.10. Immunoprecipitation and Western Blotting

Cell samples were lysed with RIPA lysis buffer containing protease inhibitor cocktail (Roche, Basel, Switzerland) for 30 min on ice and centrifuged at 13,000 rpm for 30 min. The supernatants were collected and the protein concentration was determined using a BCA kit (Thermo Scientific). For immunoprecipitation, equal amounts of lysate were incubated with anti-FLAG M2 affinity gels (A2220, Sigma) and anti-FLAG antibody overnight at 4 °C. Then, the beads were washed three times with TBST, and the immunoprecipitated proteins were analyzed by Western blotting. The protein samples were separated on 10% gel, and then the proteins were transferred onto PVDF membranes (Merck Millipore, Billerica, MA, USA). Signals of Western blotting were detected using the ImageQuant LAS-4000 mini system.

### 2.11. Sliver Staining and Mass Spectrometry

Cellular extracts from KYSE30 cells stably overexpressing empty vector and Flag-USP39 were incubated with anti-Flag M2 affinity gel (A2220, Sigma) and eluted with a 1× loading buffer. The 1× loading buffer was purchased from CoWin Biosciences (CW0027S). The proteins were collected and resolved on 10% sodium dodecyl sulfate–polyacrylamide gel electrophoresis. Gel was silver-stained by PierceTM Silver Stain for mass spectrometry (24600, Thermo Scientific) according to the manufacturer’s protocol. The proteins were identified with a gel-based liquid chromatography–tandem mass spectrometry (Gel-LS-MS/MS) approach (Beijing Qinglian Biotech Co., Ltd., Beijing, China). Mass spectra were analyzed using MaxQuant software (version 1.5.3.30) with the UniProtKB/Swiss-Prot human database.

### 2.12. RNA Isolation and qPCR

The total RNA from cells was extracted with TRIzol reagent (Invitrogen, Grand Island, NY, USA), and 1 μg of RNA was subjected to reverse transcription using a FastKing RT Kit (Tiangen, Beijing, China). Quantitative real-time PCR (qRT-PCR) was performed using the StepOneplus Real-Time PCR system (Applied Biosystems, Foster City, CA, USA) and the PowerUp SYBR Green Master Mix (Applied Biosystems) to determine the mRNA expression levels of genes. The relative expression of genes was calculated using the $2^{-∆∆Ct}$ method. The relative expression levels were normalized by the expression of the housekeeping gene GAPDH.

### 2.13. RNA Immunoprecipitation Sequencing and Data Analysis

KYSE410 cells with ectopically expressed Flag-tagged USP39 were collected for the RIP assay, which was performed using the Magna RIP Kit (Merck Millipore, Billerica MA, USA) following the manufacturer’s instructions. The cDNA libraries were constructed for each pooled RNA sample using the NEBNext® Ultra™ Directional RNA Library Prep Kit for Illumina according to the manufacturer’s instructions. Subsequently, the libraries were purified and sequenced by Illunima Noveseq 6000. The RIP-Seq clean reads were aligned to the human genome by Tophat2. After removing multiple mapped and duplicated reads, we used MACS2 to call USP39 binding peaks. 

### 2.14. RNA-Seq and Data Analysis

RNA-Seq libraries were prepared using the NEBNext Ultra RNA Library Prep Kit for Illumina (New England Biolabs, MA, USA) and were validated using an Agilent 2100 Bioanalyzer (Agilent Technologies, Santa Clara, CA, USA). Libraries were sequenced on a HiSeq 4000 sequencer (Illumina, CA, USA). After obtaining the raw reads from the sequencing platform, we used cutadapt (version 1.7.1) and the FASTX Toolkit (Version 0.0.13) to remove adapters and low-quality bases (30% bases quality less than 20). The clean reads were mapped to the human genome (GRCH38) by TopHat2, which allowed no more than 4 mismatches. The uniquely mapped reads were obtained to calculate the read number and fragments per kilobase and per million (FPKM) values for each gene. We used edgeR software to analyze the differentially expressed genes (DEGs) with a false discovery rate (FDR) of <0.01 and a fold change of 2 [23,24]. The alternative splicing events (ASEs) and regulated alternative splicing events (RASEs) were defined and quantified using rMATs pipeline. The RASE ratio was calculated by the ratio of the alternatively spliced reads and the constitutively spliced reads with a *p*-value of <0.05 and a RASE ratio of >0.2 [25].

### 2.15. GO and KEGG Analysis

Gene Ontology (http://www.geneontology.org/, accessed on 1 Sepetember 2019) and KEGG databases were used for the functional enrichment analysis of differentially enriched peaks/differentially expressed genes or differentially binding proteins pulled down by Flag-tagged USP39, respectively. A Fisher’s exact test was applied to define the enrichment degree.

### 2.16. Statistical Analysis

The statistical analyses and graphs were produced by GraphPad Prism version 8.2.0 (San Diego, CA, USA). All statistical analyses were considered statistically significant with the differences *p* ≤ 0.05. The data were similar in at least three biologically independent experiments.

## 3. Result

### 3.1. Identification of DUBs That Affect Tumor Cell Proliferation by siRNA Screening Library

To systematically identify potential DUBs that affect tumor cell proliferation, we depleted the expression of 98 human DUBs, respectively, in esophageal cancer cell line KYSE30 and detected the influence on the proliferation of tumor cells. Among the tested genes from the DUB library, we found that PSMD14 and USP39 significantly affect the proliferation of KYSE30 (Figure 1A). We preliminarily selected PSMD14 and USP39 in KYSE30 as our candidate genes. Next, we designed the shRNA of the two genes to investigate the function in tumor development and growth (Figure 1B). Xenograft experiments were performed using KYSE30 cells expressing shRNA targeting PSMD14 or USP39. The deletion of PSMD14 and USP39 significantly inhibited tumor growth (Figure 1C,D). The role of PSMD14 in ESCC has been thoroughly proved in our previous studies [26]. Finally, we focused on the function of USP39 in esophageal cancer. We examined the protein level of USP39 in 199 ESCC tissues using immunochemistry staining. We found that the high expression of USP39 correlated strongly with shorter overall and progression free survival (Figure 1E,F). Taken together, we suggest that USP39 may act as an oncogene in ESCC.

### 3.2. USP39 Promotes Esophageal Cancer Cell Proliferation, Migration, and Invasion

To further investigate the function of USP39 in ESCC, we detected the expression level of USP39 in ESCC cell lines (Figure 2A). We stably overexpressed USP39 in KYSE30 and KYSE410 cells or deleted in KYSE450 and KYSE30 cells (Figure 2B; Appendix A). Cell Counting Kit-8 (CCK-8) assays showed that USP39 overexpression promoted proliferation of KYSE30 and KYSE410 cells (Figure 2C; Appendix A), whereas the depletion of USP39 significantly inhibited proliferation of KYSE450 and KYSE30 cells (Figure 2D; Appendix A). Colony formation assays further verified the effect of USP39 on cell proliferation (Figure 2E,F; Appendix A). Consistent with the results in vitro, overexpression of USP39 significantly promoted tumor growth, whereas the depletion of USP39 inhibited tumor growth compared with the corresponding control group (Figure 2G,H), suggesting that USP39 promotes esophageal cancer cell growth. Subsequently, we examined the effects of USP39 on cell migration and invasion. The transwell assay showed that the overexpression of USP39 promoted the migrative and invasive capabilities of esophageal cancer cells, while the suppression of USP39 exerted the opposite effect (Appendix A). Altogether, these results prove that USP39 may exert a tumor-promoting property in ESCC.

### 3.3. USP39 Drives Chemoresistance in Esophageal Cancer via Inhibition of Cell Apoptosis

Chemoresistance remains a main clinical obstacle in the treatment of ESCC patients. The strong correlation between high expressions of USP39 with poor prognosis prompted us to investigate the roles of USP39 in the chemoresistance of esophageal cancer cells. Flow cytometric analysis showed that the ectopic expression of USP39 had no effect on cell apoptosis, but reduced the apoptotic rate of esophageal cancer cells with cisplatin (DDP) treatment (Figure 3A; Appendix A). In contrast, knockdown of USP39 had a slight effect on cell apoptosis without DDP treatment and obviously increased DDP-induced cell apoptotic rate (Figure 3B; Appendix A). Immunoblotting analysis further verified the effect of USP39 on the apoptosis markers expression. The expression level of cleaved PARP and Caspase-3 was significantly increased when treated with DDP, and the ectopic expression of USP39 dramatically inhibited the activation of PARP and Caspase-3 in KYSE30 cells (Figure 3C). These results illustrate that USP39 triggers chemoresistance in ESCC by inhibiting cell apoptosis.

### 3.4. USP39 Interacts with Several Spliceosome Components

To elucidate the molecular mechanisms underlying USP39-driven malignant phenotypes in ESCC, we performed immunoprecipitation and mass spectrometry to interrogate the USP39 interactome in KYSE30 cells that stably overexpressed Flag-tagged USP39 (Figure 4A,B). Gene Ontology term enrichment analysis revealed that the potential interacting proteins with USP39 were involved in "spliceosome mediated mRNA splicing", which included a total of 17 proteins (36.2%), such as EFTUD2, PRPF3, PRPF31, and PRP6. This was followed by "RNA splicing" and "mRNA processing" with 13 (27.7%) and 12 (25.5%) proteins involved, respectively (Figure 4C). In the subsequent analysis of the KEGG signaling pathway, 14 (30%) proteins were enriched in the splicing signaling pathway, 7 (15%) proteins were enriched in the ribosome signaling pathway, and the remaining proteins were not enriched in any specific signaling pathway (Figure 4D). Then, we validated the interaction between USP39 and the spliceosome components proteins identified by mass spectrometry using coimmunoprecipitation (Co-IP) and Western blot. EFTUD2, PRPF3, SART1, DDX23, and hnRNPU were verified as USP39-interacting partners. Our data suggest that USP39 may act as a regulator of RNA splicing to promote the malignant phenotypes of ESCC.

### 3.5. USP39 Affects Alternative Splicing in ESCC

To explore the USP39-regulated alternative splicing events, we next performed RIP-seq using an anti-USP39 antibody in KYSE410 cells, which stably overexpressed USP39. We identified 3981 binding peaks and the sequencing results show that most of the USP39 binding sites mapped to exons, as illustrated in Figure 5A. Next, we conducted GO analysis of genes bound by USP39 (Figure 5B). The results show that the phosphorylation and chromatin organization pathways were enriched. To further identify the alternative splicing events (AS), we performed RNA-Seq in USP39 silenced cells and control cells. In total, 3496 AS events were identified. Go function analysis revealed that USP39-regualted AS events are involved in cell cycle and chromatin organization pathways (Figure 5C). The integrated analysis of RIP-Seq and RNA-Seq showed that USP39 regulates hundreds of AS events by direct RNA binding (Figure 5D). Furthermore, we validated the AS events regulated by USP39 by RT-PCR and showed that USP39 regulated the alternative splicing of ESPR2, FOXK2, CCT7, HNRNPC, and NCOA4 (Figure 5E). The results suggested that USP39 affects alternative splicing in ESCC.

## 4. Discussion

Previous studies have reported that USP39 contributes to cancer progression in various human tumor types and acts as a splicing factor. In ovarian cancer, USP39 promotes the proliferation and invasion of cancer cells by facilitating efficient splicing of the oncogenic transcription factor HMGA2 [8]. Moreover, USP39 promotes ESCC tumorigenesis through regulating Rictor pre-mRNA splicing [27]. In our study, we further demonstrate the oncogenic function of USP39 in ESCC. In functional experiments, we demonstrated that USP39 not only promotes the proliferation of cancer cells but also influences cell migration, invasion, and chemoresistance in ESCC. Based on the analysis of mass spectrometry, we found that USP39 interacts with several spliceosome components, such as EFTUD2, PRPF3, SART1, DDX23, and hnRNPU. EFTUD2, PRPF3, and SART1 are reported to be a component of the U4/U6-U5 tri-snRNP complex [12,28,29] and the USP39 is an essential factor for the recruitment of U4/U6-U5 tri-snRNP. DDX23 is required for spliceosome B complex formation [30]. Through the analysis of the GO database, we found that the proteins most frequently enriched biological processes associated with mRNA splicing and RNA processing. Altogether, our results further support the idea that USP39 acts as a splicing factor in promoting the progression of ESCC.

In order to identify the alternative splicing events regulated by USP39, we performed RNA-seq and RIP-seq approaches. With the combined analysis of RNA-seq and RIP-seq, we identified that USP39 affects the splicing events of multiple members of the MCM family, such as MCM3, forming a highly conserved DNA-binding protein family which affects the initiation and regulation of DNA replication [31,32]. The overexpression of MCM3 in tumor tissues predicts poor survival in hepatocellular carcinoma [33]. The expression of MCM3 has also been associated with poor prognoses in malignant glioma and medulloblastoma [34,35]. We also validated changes in *ESPR2, FOXK2, CCT7, HNRNPC* and *NCOA4* splicing by RT-PCR in USP39-silenced cells. Among them, HNRNPC, a m6A regulator has been reported to be significantly correlated with worse outcomes and immune infiltration levels in esophageal cancer [36]. In future studies, we will further explore and validate the functional alternative splicing events regulated by USP39 in ESCC.

Taken together, our study demonstrates that USP39 serves as an oncogenic factor in ESCC through promoting tumor proliferation in vivo and in vitro, improving invasion and migration of cancer cells and inhibiting the cell apoptosis with the treatment of DDP. We also provide evidence that USP39 interacts with several spliceosome components by IP-MS. Finally, the combined analysis of RNA-seq and RIP-seq demonstrates that USP39 regulated hundreds of the alternative splicing events. USP39 could provide an important basis for the diagnosis and therapy of ESCC.

## 5. Conclusions

In summary, our findings reveal USP39 promotes malignant phenotypes of ESCC cells as a splicing factor and identify USP39 as a new biomarker of poor prognosis and a candidate therapeutic target in ESCC.

## Figures and Tables

**Figure 1 genes-13-00819-f001:**
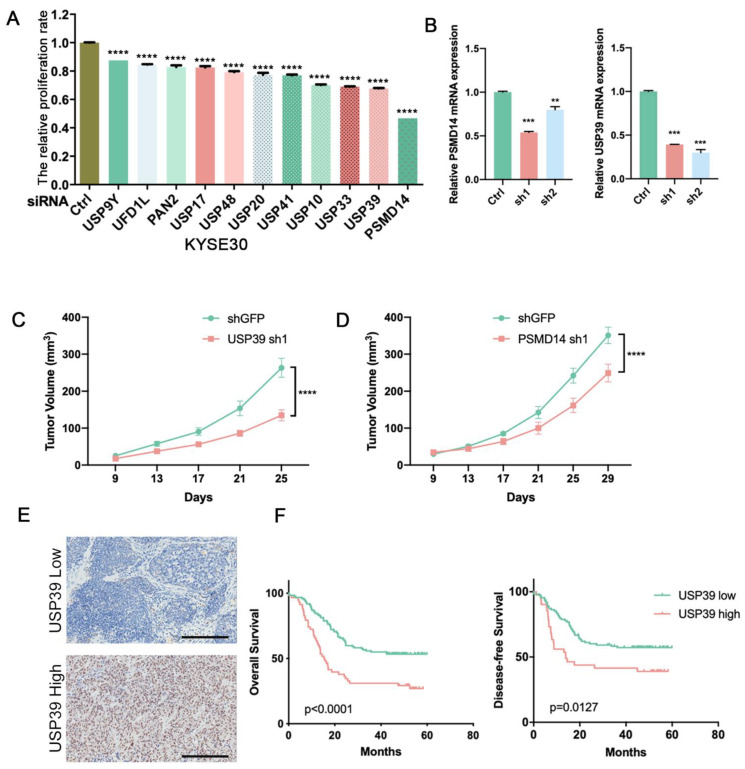
Identification of the DUB that promotes the proliferation of esophageal cancer cells. (**A**) Each of the 98 human DUB genes was suppressed in KYSE30 cells with pooled siRNAs and the cell proliferation was detected by the CCK-8 assay. (**B**) Knockdown efficiency of PSMD14 and USP39 in KYSE30 cells was examined by qRT-PCR. (**C**) Tumor volumes of USP39 silenced KYSE30 cells and control cells. *n* = 7 mice per group. (**D**) Tumor volumes of PSMD14 silenced KYSE30 cells and control cells. *n* = 7 mice per group. (**E**) Representative images of IHC staining of USP39 in USP39 high expression or USP39 low expression cases. Scale bars, 200 μm. (**F**) Kaplan–Meier analysis of ESCC patients’ overall survival and disease-free survival grouped by low or high expression of USP39 by the log rank test. Data in A, B, C, D are shown as the mean ± S.D. Data were analyzed by one-way ANOVA with Bonferroni correction (**A**,**B**) or two-way ANOVA with Bonferroni correction (**C**,**D**). ** *p* < 0.01, *** *p* < 0.001, **** *p* < 0.0001.

**Figure 2 genes-13-00819-f002:**
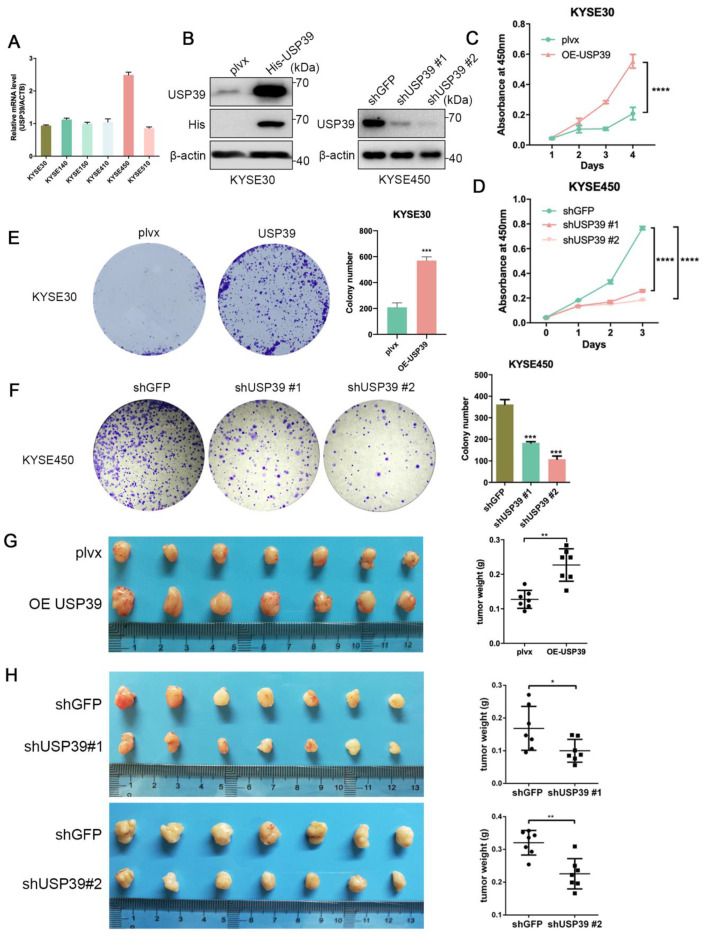
USP39 promotes the proliferation of ESCC cells *in vivo* and *in vitro*. (**A**) Quantitative real-time PCR detection of USP39 expression in ESCC cell lines. (**B**) Overexpression and knockdown efficiency of USP39 in KYSE30 and KYSE450 was determined by Western blot. (**C**,**D**) Cell proliferation detected by the CCK-8 assay after USP39 overexpression (**C**) or knockdown (**D**) in KYSE30 cells and KYSE450 cells, respectively. (**E**,**F**) Cell proliferation detected by colony formation assays after USP39 overexpression (**E**) or knockdown (**F**) in KYSE30 and KYSE450 cells, respectively. *n* = 3 per group. (**G**,**H**) Images of xenograft tumors, tumor weight in mice subcutaneously injected with ESCC cells overexpression (**G**) or knockdown (**H**) of USP39. *n* = 7 mice per group. Data in C-H represent mean ± SD. Data were analyzed by two-way ANOVA (**C**,**D**) with Bonferroni correction or unpaired two-tailed Student’s *t*-test (**E**–**H**). ns = no significant, * *p* < 0.05, ** *p* < 0.01, *** *p* < 0.001, **** *p* < 0.0001.

**Figure 3 genes-13-00819-f003:**
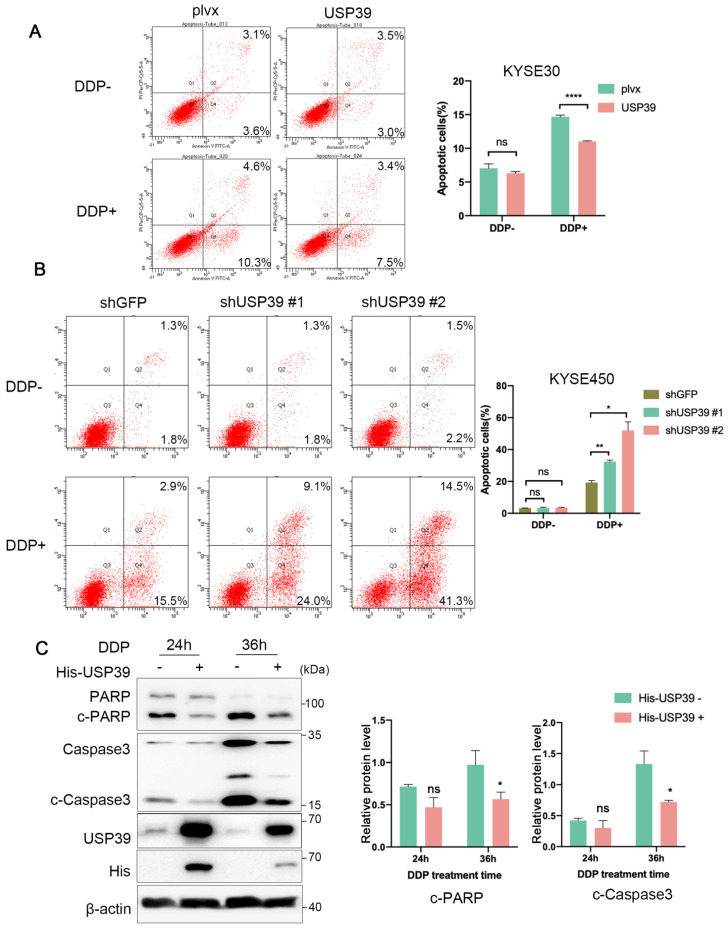
USP39 promotes chemoresistance in ESCC cells. (**A**) Cell apoptosis detected by the Annexin V-PI assay in USP39-overexpressed KYSE30 and control cells with or without DDP (14 µg/mL) treatment. *n* = 3. (**B**) Cell apoptosis detected by the Annexin V-PI assay after USP39 knockdown in KYSE450 cells with or without DDP (7 µg/mL) treatment. *n* = 3. (**C**) Representative immunoblotting analysis of the cleaved PARP and Caspase3 expression in USP39-overexpressed KYSE30 cells with DDP treatment. The blot was analyzed by Image J. Relative protein levels were normalized to β-actin. Data in A, B, and C represent mean ± S.D and were analyzed by unpaired two-tailed Student’s *t*-test. ns = no significant, * *p* < 0.05, ** *p* < 0.01, **** *p* < 0.0001.

**Figure 4 genes-13-00819-f004:**
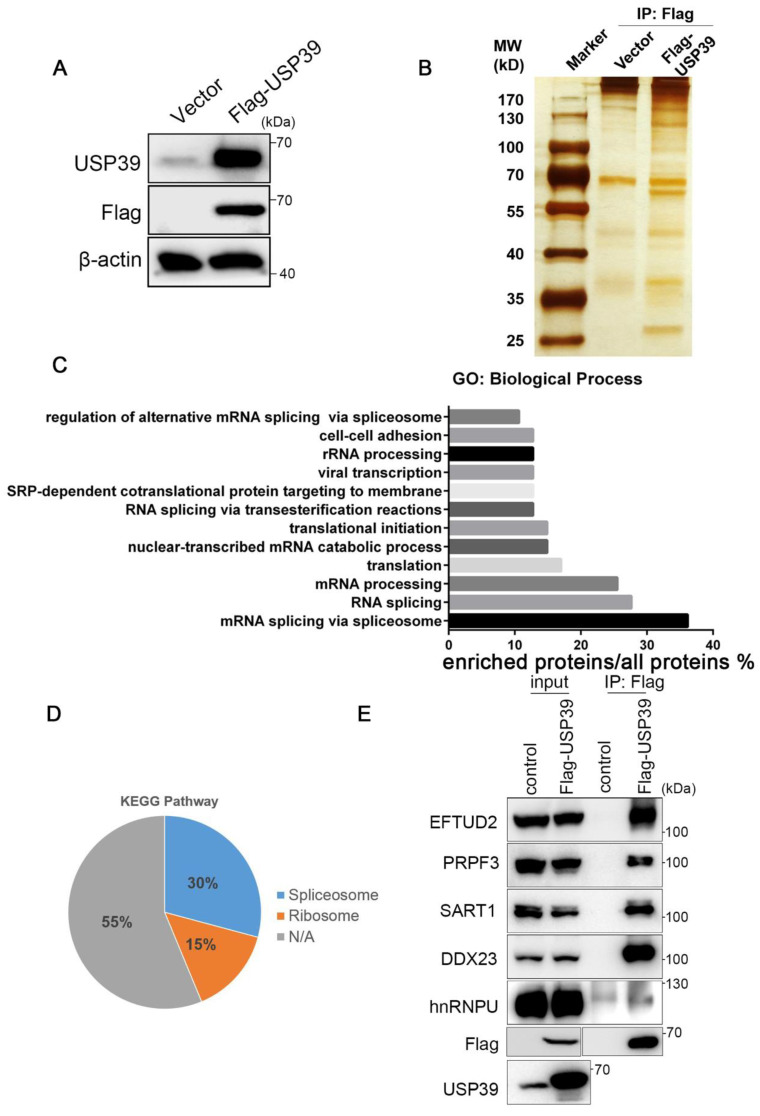
USP39 interacts with several spliceosome components. (**A**) Overexpression efficiency of USP39 in KYSE30 cells was determined by Western blot. (**B**) The USP39 stable expression clone was established. Proteins that interacted with USP39 were purified from KYSE30 cells expressing Flag-tagged USP39 or vector control. (**C**) GO enrichment analysis of proteins pulled down by Flag antibody from KYSE30 cells expressing Flag-tagged USP39. (**D**) KEGG pathway analysis of proteins pulled down by Flag antibody from KYSE30 cells expressing Flag-tagged USP39. (**E**) The interaction between USP39 and core spliceosome factors was validated by Co-IP in KYSE30 cells.

**Figure 5 genes-13-00819-f005:**
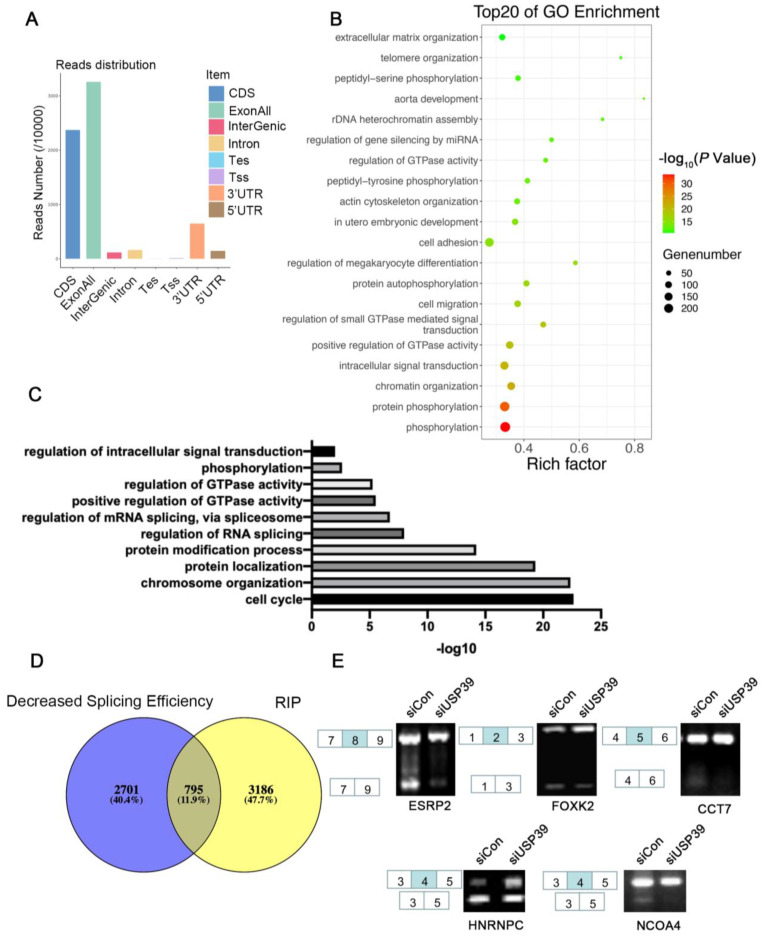
Identification of USP39-regulated splicing events by RNA-seq and RIP-seq. (**A**) Read numbers of USP39 binding peaks based on RIP-seq data. Coding sequence (CDS), transcription start site (Tss), transcription end site (Tes), exon, intron, 3’ untranslated region (3’UTR), and 5’ untranslated region (5’UTR) were extracted by human genome gff3 file from the Ensembl database. (**B**) Bubble diagram showing the results of GO enrichment analysis of genes bound by USP39. (**C**) GO enrichment analysis of genes that presented AS events regulated by USP39. (**D**) The overlap in transcripts identified as being bound by USP39 (from RIP-seq) and different splicing efficiency upon USP39 knockdown (from RNA-seq). (**E**) The AS events regulated by USP39 were detected by RT-PCR. The primer lists are shown in Appendix A.

## Data Availability

Files were downloaded from the Genome Sequence Archive (GSA) in the BIG Data Center (http://bigd.big.ac.cn/gsa), the Beijing Institute of Genomics (BIG), the Chinese Academy of Sciences, with the BioProject number PRJCA009047.

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
