# Peer review of "The Deubiquitinase USP39 Promotes Esophageal Squamous Cell Carcinoma Malignancy as a Splicing Factor"

_genes, 2022, doi:10.3390/genes13050819_

Round 1
Reviewer 1 Report
In this Article, the authors identified by functional screening the deubiquitinase enzyme ubiquitin specific protease 39 (USP39) in esophageal squamous cell carcinoma (ESCC), an aggressive epithelial malignancy. High expression of USP39 promoted cancer cell growth and chemoresistance in ESCC and correlated with shorter overall survival and progression-free survival of ESCC patients. Analysis of RNA-seq and RNA immunoprecipitation (RIP) sequencing suggest that USP39 regulates alternative splicing, in particular the splicing of minichromosome maintenance complex component 3 (MCM3), in ESCC.
Although the study is potentially interesting, the authors must better explain the rationale for their work and adhere to more rigorous experimental standards. In particular, immunohistochemistry (IHC) results should be scored based on the number of immunopositive cells, not on signal intensity, for consistent results. Silencing with more than one short hairpin RNA (shRNA) and rescue experiments are needed to demonstrate the specificity of the target silencing throughout the experimental section (Fig. 1-5).
Line 46: Elaborate on the known mechanisms and biological significance of alternate splicing in human ESCC. See Wu et al. and Dlamini et al. in the list of references below, for example.
Line 51: Describe the minichromosome maintenance complex (MCM) components, in particular MCM3, earlier in this introduction. Importantly, state in sufficient detail your research hypothesis and the rationale for your work.
Line 61-62: Where were USP39, USP39 shRNA and PSMD14 shRNA obtained?
Line 81-82: What were the 4 siRNA sequences in the siRNA pool?
Line 93: Explain the H-score and provide a pertinent reference.
Line 105: Were the KYSE30 cells injected s.c. in the mouse flank?
Line 158: Describe the demographic and pathological (e.g., primary, recurrent or metastatic, stage, grade, etc.) composition of the 199 ESCC samples.
Line 161-163: Grading of USP39 IHC results should have been based on the number of tumor cells immunopositive for USP39, not the signal intensity because the latter is affected by differences in IHC conditions among different batches of IHC assays of ESCC specimens.
Line 167-169: Describe the statistical analyses performed.
Line 177: Why was the siRNA screen performed in only one ESCC cell line (KYSE30)? How generalizable is this single result?
Line 181: Silencing of USP39 or PSMD14 with only one shRNA each is inadequate. To control for the possibility of off-target effects, experiments should be repeated using 2 or 3 different shRNAs for each target, and rescue experiments should be conducted to determine whether overexpression of the target can overturn the silencing effect of its shRNAs.
Line 185-186: Define the signal intensity for USP39 high and low. How many cases of the 199 cases fell into each group?
Line 191-192: Why did you switch to SW527 cells for these experiments?
Line 196: Because experiments in Fig. 1A-D involved multiple groups or time points, one-way ANOVA should be performed followed by post-hoc test (e.g., Tukey) for comparisons.
Line 202-204: Same comments regarding overexpression and silencing as for Line 181.
Line 230, 240: Is DDP cisplatin? This should be defined and described under Methods.
Line 276: Which criteria did you use to choose to focus on MCM3 for further study?
Line 283: Define CDS, Tes, Tss, UTR3 and UTR5. Also, Rich factor.
Line 285: Define what were considered to be differences in expression (using edgeR software) and differences in splicing efficiency. How was splicing efficiency measured experimentally?
Line 287-288: Define the units of expression level of USP39 and MCM3.
Line 294-295: Explain how your present findings relate to your previous paper (Ref. 24) about the tumorigenic effect of USP39 via Rictor pre-mRNA splicing in ESCC?
Line 310-316: Other MCM proteins (MCM2, MCM4, MCM6, MCM7) have been identified in ESCC (see references by Li, Yu, Choy and Huang below). Is the splicing of these MCM mRNAs also affected by USP39?
The list of references is not entirely up-to-date. Consider the following references:
- Wu Q, et al. The landscape and biological relevance of aberrant alternative splicing events in esophageal squamous cell carcinoma. Oncogene. 2021 Jun;40(24):4184-4197. doi: 10.1038/s41388-021-01849-8.
- Dlamini Z, et al. Prognostic alternative splicing signatures in esophageal carcinoma. Cancer Manag Res. 2021 Jun 4;13:4509-4527. doi: 10.2147/CMAR.S305464.
- Li X, et al. Minichromosome maintenance 6 complex component identified by bioinformatics analysis and experimental validation in esophageal squamous cell carcinoma. Oncol Rep. 2020 Sep;44(3):987-1002. doi: 10.3892/or.2020.7658.
- Yu J, et al. Knockdown of Minichromosome Maintenance proteins inhibits foci forming of mediator of DNA-Damage Checkpoint 1 in response to DNA damage in human esophageal squamous cell carcinoma TE-1 cells. Biochemistry (Mosc). 2016 Oct;81(10):1221-1228. doi: 10.1134/S0006297916100205.
- Choy B, et al. MCM4 and MCM7, potential novel proliferation markers, significantly correlated with Ki-67, Bmi1, and cyclin E expression in esophageal adenocarcinoma, squamous cell carcinoma, and precancerous lesions. Hum Pathol. 2016 Nov;57:126-135. doi: 10.1016/j.humpath.2016.07.013.
- Huang B, et al. Potential role of minichromosome maintenance protein 2 as a screening biomarker in esophageal cancer high-risk population in China. Hum Pathol. 2011 Jun;42(6):808-16. doi: 10.1016/j.humpath.2010.04.022.
Author Response
Response to Reviewer 1 Comments
In this Article, the authors identified by functional screening the deubiquitinase enzyme ubiquitin specific protease 39 (USP39) in esophageal squamous cell carcinoma (ESCC), an aggressive epithelial malignancy. High expression of USP39 promoted cancer cell growth and chemoresistance in ESCC and correlated with shorter overall survival and progression-free survival of ESCC patients. Analysis of RNA-seq and RNA immunoprecipitation (RIP) sequencing suggest that USP39 regulates alternative splicing, in particular the splicing of minichromosome maintenance complex component 3 (MCM3), in ESCC.
Although the study is potentially interesting, the authors must better explain the rationale for their work and adhere to more rigorous experimental standards. In particular, immunohistochemistry (IHC) results should be scored based on the number of immunopositive cells, not on signal intensity, for consistent results. Silencing with more than one short hairpin RNA (shRNA) and rescue experiments are needed to demonstrate the specificity of the target silencing throughout the experimental section (Fig. 1-5).
- Line 46: Elaborate on the known mechanisms and biological significance of alternate splicing in human ESCC. See Wu et al. and Dlamini et al. in the list of references below, for example.
Response 1: We appreciate the reviewer’s comment. We have supplemented the description on the mechanisms and biological significance of alternate splicing in the paragraph 4 of the introduction section.
- Line 51: Describe the minichromosome maintenance complex (MCM) components, in particular MCM3, earlier in this introduction. Importantly, state in sufficient detail your research hypothesis and the rationale for your work.
Response 2: We appreciate the reviewer’s comment, and reviewer 2# & 3# also raised some concerns about MCM3. Currently, we only provide the preliminary evidence show that USP39 may regulate the alternate splicing. And no evidence to elucidate the underlying mechanism of USP39 regulating MCM3 splicing and prove that USP39 promotes ESCC malignancy by affecting the alternative splicing events of MCM3. So we decided to remove the results of MCM3 in this manuscript. In the future, we will further explore the functional alternative splicing events regulated by USP39.
- Line 61-62: Where were USP39, USP39 shRNA and PSMD14 shRNA obtained?
Response 3: We are sorry for not providing the detailed information for vector construction, we have complemented the description in the Materials and Methods section.
- Line 81-82: What were the 4 siRNA sequences in the siRNA pool?
Response 4: We are sorry for not providing siRNA sequences of siUSP39. The 4 siRNA sequences targeting USP39 in the siRNA pool is as follows:
siUSP39-1: GAUCAUCGAUUCCUCAUUG
siUSP39-2: CAUAUGAUGGUACCACUUA
siUSP39-3: GAGGAUAUCACGUAUGUGU
siUSP39-4: GCUUCCAGCUUACCAAGUU
The siRNA sequences targeting USP39 are also included in Supplementary Table 1.
- Line 93: Explain the H-score and provide a pertinent reference.
Response 5: We are sorry we did not clearly describe the methods on IHC analysis. IHC results was scored with H Score, a final score was by multiplying the scores of“percentage of USP39-positive cells”and staining intensity (assigned on 4 level, 0, negative; 1, weak staining; 2, mild staining; 3,strong staining). The staining intensity were evaluated by two pathologists independently. The description and a pertinent reference have been complemented in the Immunohistochemistry Staining section.
- Line 105: Were the KYSE30 cells injected s.c. in the mouse flank?
Response 6: In the xenograft transplantation experiments, the KYSE30 cells were subcutaneously injected in the right armpit of mice.
- Line 158: Describe the demographic and pathological (e.g., primary, recurrent or metastatic, stage, grade, etc.) composition of the 199 ESCC samples.
Response 7: The information of the 199 ESCC samples has been added in the Supplementary Table 2
- Line 161-163: Grading of USP39 IHC results should have been based on the number of tumor cells immunopositive for USP39, not the signal intensity because the latter is affected by differences in IHC conditions among different batches of IHC assays of ESCC specimens.
Response 8: Please see the Response 5, above. We analyzed the IHC staining by H-score, a standardized IHC scoring method.
- Line 167-169: Describe the statistical analyses performed.
Response 9: Please see the Response 5&8, above.
- Line 177: Why was the siRNA screen performed in only one ESCC cell line (KYSE30)? How generalizable is this single result?
Response 10: We appreciate the reviewer’s comment. Actually, we performed screening of DUB library in both KYSE30 and SW527 cells with high transfection efficiency. Based on the results in KYSE30 cells, we selected USP39 and PSMD14 for candidate genes. Consistently, knockdown of USP39 and PSMD14 exerted the inhibitory effect on cell proliferation in SW527 cells (See Response Figure 1).
Response Figure 1. Each of the 98 human DUB genes was transfected in SW527 cells with pooled siRNA and the cell proliferation was detected by CCK8 assay. Data were analyzed by one-way ANOVA with Bonferroni correction. **** p < 0.0001. Please see the attached file.
- Line 181: Silencing of USP39 or PSMD14 with only one shRNA each is inadequate. To control for the possibility of off-target effects, experiments should be repeated using 2 or 3 different shRNAs for each target, and rescue experiments should be conducted to determine whether overexpression of the target can overturn the silencing effect of its shRNAs.
Response 11: We respectfully agree with the reviewer’s comment. We would like to clarify that we used one shRNA to validate the results obtained by siRNA transfectionin the screening stage (As shown in Figure 1C&1D). And we used two independent shRNAs targeting USP39 for further functional validation (As shown in Figure 2B Right panel, 2D, 2F&2H). Meanwhile, we also overexpressed USP39 to perform functional assays to further confirm the effect on the malignant phenotype of USP39.
- Line 185-186: Define the signal intensity for USP39 high and low. How many cases of the 199 cases fell into each group?
Response 12: We appreciate the reviewer’s comment. The signal intensity for USP39 was divided in four level. 0, negative; 1, weak staining; 2, mild staining; 3, strong staining. We divided the patients into two groups based on the average of H-score. 130 cases were divided into low-expression USP39 group and 69 cases were divided into high-expression USP39 group.
- Line 191-192: Why did you switch to SW527 cells for these experiments?
Response 13: We are sorry for misspelling KYSE30 to SW527 and we have corrected it in Figure 1 legends.
- Line 196: Because experiments in Fig. 1A-D involved multiple groups or time points, one-way ANOVA should be performed followed by post-hoc test (e.g., Tukey) for comparisons.
Response 14: Thanks for this reviewer’s suggestion. We have checked and revised the statistical methods in Fig. 1A-D. In Fig.1A-B, the data were analyzed by one-way ANOVA with Bonferroni. In Fig. 1C-D, the data were analyzed by two-way ANOVA with Bonferroni.
- Line 202-204: Same comments regarding overexpression and silencing as for Line 181.
Response 15: We appreciate the reviewer’s comment. Please see the response 11, above.
- Line 230, 240: Is DDP cisplatin? This should be defined and described under Methods.
Response 16: Yes, cisplatin is abbreviated as DDP and we have defined cisplatin (DDP) where it firstly appeared.
- Line 276: Which criteria did you use to choose to focus on MCM3 for further study?
Response 17: We appreciate the reviewer’s comment. Previously, we acquired MYOF, MCM3 and PDPR by integrately analyzing the data of RIP-seq and RNA-seq. We next combined the expression correlation analysis with literatures searching to focus on MCM3. Currently, we decided to remove the MCM3 from the results section since the data are not adequate for selecting and verifying MCM3 as a USP39-regulated downstream functional target (see Response 2, above).
- Line 283: Define CDS, Tes, Tss, UTR3 and UTR5. Also, Rich factor.
Response 18: CDS is coding sequence, Tes is transcription end site, Tss is transcription start site, 3’UTR is 3’-untranslated region, 5’UTR is 5’-untranslated region. And Exon, Intron, 3’UTR and 5’UTR was extracted by human genome gff3 file from ensembl database (GRCH38).
- Line 285: Define what were considered to be differences in expression (using edgeR software) and differences in splicing efficiency. How was splicing efficiency measured experimentally?
Response 19: We performed bioinformatics analysis as previously described (Liu et al, Nat Commun, 2020). Differentially expressed genes (DEGs) were determined using edgeR software with a false discovery rate (FDR) <0.01 and a fold change of 2. The alternative splicing events (ASEs) and regulated alternative splicing events (RASEs) were defined and quantified using rMATs pipeline. The RASE ratio was calculated by the ratio of the alternatively spliced reads and the constitutively spliced reads with p-value < 0.05 and RASE ratio > 0.2. We have added the description in the Methods section.
- Line 287-288: Define the units of expression level of USP39 and MCM3.
Response 20: The units of expression level of USP39 and MCM3 was obtained from expression dataset from ESCC patients (GSE161533).
- Line 294-295: Explain how your present findings relate to your previous paper (Ref. 24) about the tumorigenic effect of USP39 via Rictor pre-mRNA splicing in ESCC?
Response 21: We appreciate the reviewer’s comment. In terms of the biological phenotypes, Zhao et al. focus on the effect by USP39 knockdown on esophageal cancer cell growth in vitro and in vivo,while we comprehensively investigated the effect on the malignant phenotypes including cell proliferation, migration, invasion and chemosensitivity etc. by USP39 overexpression and knockdown. Mechanistically, Zhao et al. identify a specific downstream target regulated by USP39, while we screened and validated the interaction between USP39 and splicesome components. We also intergrately analyzed RIP-Seq and RNA-Seq data to systematically identify the alternative splicing events regulated by USP39 at the genome-wide scale. Altogether, the two studies complement each other to demonstrate that USP39 exerts a tumor-promoting property by regulating mRNA splicing in ESCC.
- Line 310-316: Other MCM proteins (MCM2, MCM4, MCM6, MCM7) have been identified in ESCC (see references by Li, Yu, Choy and Huang below). Is the splicing of these MCM mRNAs also affected by USP39?
Response 22: We appreciate the reviewer’s comment. RIP-Seq showed that USP39 binds directly to the MCM2, MCM3, MCM4, MCM6, MCM7 and MCM9 etc mRNA, and RNA-Seq showed that USP39 regulates the splicing of MCM3, MCM8 and MCM9 etc. The results demonstrate that USP39 may be involved in the splicing of multiple members of MCM family, suggesting that it is not sufficient to select MCM3 for further investigation. Combined with Reviewer 1# and 2# concerns, we decided to remove MCM3 results.
- The list of references is not entirely up-to-date. Consider the following references:
- Wu Q, et al. The landscape and biological relevance of aberrant alternative splicing events in esophageal squamous cell carcinoma. Oncogene. 2021 Jun;40(24):4184-4197. doi: 10.1038/s41388-021-01849-8.
- Dlamini Z, et al. Prognostic alternative splicing signatures in esophageal carcinoma. Cancer Manag Res. 2021 Jun 4;13:4509-4527. doi: 10.2147/CMAR.S305464.
- Li X, et al. Minichromosome maintenance 6 complex component identified by bioinformatics analysis and experimental validation in esophageal squamous cell carcinoma. Oncol Rep. 2020 Sep;44(3):987-1002. doi: 10.3892/or.2020.7658.
- Yu J, et al. Knockdown of Minichromosome Maintenance proteins inhibits foci forming of mediator of DNA-Damage Checkpoint 1 in response to DNA damage in human esophageal squamous cell carcinoma TE-1 cells. Biochemistry (Mosc). 2016 Oct;81(10):1221-1228. doi: 10.1134/S0006297916100205.
- Choy B, et al. MCM4 and MCM7, potential novel proliferation markers, significantly correlated with Ki-67, Bmi1, and cyclin E expression in esophageal adenocarcinoma, squamous cell carcinoma, and precancerous lesions. Hum Pathol. 2016 Nov;57:126-135. doi: 10.1016/j.humpath.2016.07.013.
- Huang B, et al. Potential role of minichromosome maintenance protein 2 as a screening biomarker in esophageal cancer high-risk population in China. Hum Pathol. 2011 Jun;42(6):808-16. doi: 10.1016/j.humpath.2010.04.022.
Response 23: We do appreciate for this reviewer’s efforts for our manuscript. We have added some recommended references to this manuscript.
In light of these improvements, we are optimistic that all of the concerns have been addressed to your satisfaction, and that you and the reviewer will consider this work appropriate for Genes. Thanks again for all of your efforts in handling our manuscript.

Reviewer 2 Report
In their work “The Deubiquitinase USP39 Promotes ESCC Malignancy via MCM3 SPlicing“, Zhu and colleagues investigate the role of USP39 in ESCC.
The rationale for the identification of genes involved in ESCC progression is solid. Significantly, it is based on a large cohort of ESCC tumor samples. Altogether these results reinforce the role of USP39 in cancer biology, as already observed by other groups.
Next, the authors try to describe the role of USP39 in chemotherapy resistance. These data are potential interestingly. However, the authors fail to link these findings to USP39 splicing activity or other USP39 biological functions.
One of the most striking effects is the complete abolition of PARP expression in USP39 overexpressing cells (Fig 3C). This result is not discussed in the paper. Since it is not observed in ovarian cancer cells overexpressing USP39, it is essential to validate this finding in additional cell lines (at least KYSE410).
The authors also confirmed the involvement of USP39 in the spliceosomal assembly, as already demonstrated by others, but, in my opinion, fail to correctly investigate the role of USP39 in alternative splicing regulation in ESCC.
The reasons behind their focus on MCM3 are not clear, and some data (validation of MCM3 splicing) does not seem robust enough. I will suggest the authors go deeply into the regulation and consequences of MCM3 splicing alterations.
The authors should not limit their attention to genes that change both splicing and expression profiles. Alternative splicing events can affect tumorigenesis by regulating the localization and activity of their targets.
Material and methods should be drastically improved. Spelling and grammar errors are present and should be addressed.
My suggestion is that the article should be extensively revised.
Major comments:
- Figure 1 legend does not reflect Figure 1 panels.
Lines 172-176 - Authors claim to have tested KYSE30 proliferation after testing a DUB library; however, in Figure 1A they show cell viability data. Does depletion of USP39 reduce cell viability or slow down the proliferation rate?
-Material and methods section should be drastically improved.
Specifically:
- cloning strategy to generate pLVX-USP39 and pSIH-H1-shUSP9/PSMD14 is not reported.
-Virus infection is poorly described.
- there is no description of the detection system for immunoblot analysis.
There is no indication of buffer composition used in the silver staining and mass spectrometry sub-section (loading buffer 1x composition should be provided).
- Primers used for qPCR assays should be included in the supplementary file.
- There is no information about how the authors performed GO and the KEGG analysis.
Line 214 – I will carefully rephrase this sentence because there is no demonstration that overexpression of USP39 can promote tumorigenesis of non-transformed cells.
Line 226-228 – It is not clear if tissue samples analyzed for USP39 protein expression by IHC derive from patients treated with chemotherapy, radiation, both, or nothing. Otherwise, it is not possible to correlate more survival with chemotherapy resistance.
- Figure 4E. The Flag panel should not be split. Moreover, I could not understand why there is a Flag signal in the IP control lane. Similarly, also USP39 pattern could be shown in the IP lanes.
- It is not clear why the authors focused on MCM3 only, the alternative splicing events that is regulated, and why there is a correlation between USP39 and MCM3 expression. Is the alternative exon encoded for a cryptic exon that leads to AS-NMD activation?
- Differences in Fig. 5D and 5E are very slight. Did the author investigate the AS event also through PCR and gel electrophoresis to discriminate among AS variants?
Minor comments:
- Gene and transcript names should be in italics (i.e. line 38 and in the figures);
- No error bars for PSMD14 condition are visible in Figure 1A.
- Cloning procedures in the material and methods section could be improved;
-Figure 3C, 4A. Protein size is missing.
-To properly evaluate Flow cytometry data, the gating strategy used for the analysis should be included.
- I will include in the main text also the common name of DPP.
-Supplementary Figure 1E. The quality of the colony formation assay images is very poor. It is hard to see colonies in shUSP39 #1 and #2 as quantified in the plot on the right.
-Figures 4A and 4b are more appropriate as supplementary panels.
- There is no information about the statistical test used in Supplementary Fig 1 and 2.
- The numerosity (n) of each group should be reported in each analysis
-Fig. 5A – A pie chart will be more informative. Also, the definition of Tes, Tss, ExonAll should be included in the figure legend.
- Fig. 5D and E. Please correct “soliced mRNA/mRNA”.
- Line 280 – It is not clear which GEO expression dataset has been used.
Author Response
Response to Reviewer 2 Comments
Comments and Suggestions for Authors
In their work “The Deubiquitinase USP39 Promotes ESCC Malignancy via MCM3 SPlicing“, Zhu and colleagues investigate the role of USP39 in ESCC.
The rationale for the identification of genes involved in ESCC progression is solid. Significantly, it is based on a large cohort of ESCC tumor samples. Altogether these results reinforce the role of USP39 in cancer biology, as already observed by other groups.
Next, the authors try to describe the role of USP39 in chemotherapy resistance. These data are potential interestingly. However, the authors fail to link these findings to USP39 splicing activity or other USP39 biological functions.
One of the most striking effects is the complete abolition of PARP expression in USP39 overexpressing cells (Fig 3C). This result is not discussed in the paper. Since it is not observed in ovarian cancer cells overexpressing USP39, it is essential to validate this finding in additional cell lines (at least KYSE410).
The authors also confirmed the involvement of USP39 in the spliceosomal assembly, as already demonstrated by others, but, in my opinion, fail to correctly investigate the role of USP39 in alternative splicing regulation in ESCC.
The reasons behind their focus on MCM3 are not clear, and some data (validation of MCM3 splicing) does not seem robust enough. I will suggest the authors go deeply into the regulation and consequences of MCM3 splicing alterations.
The authors should not limit their attention to genes that change both splicing and expression profiles. Alternative splicing events can affect tumorigenesis by regulating the localization and activity of their targets.
Material and methods should be drastically improved. Spelling and grammar errors are present and should be addressed.
My suggestion is that the article should be extensively revised.
Major comments:
- Figure 1 legend does not reflect Figure 1 panels.
Response 1: We have carefully revised the Figure 1 legends.
- Lines 172-176 - Authors claim to have tested KYSE30 proliferation after testing a DUB library; however, in Figure 1A they show cell viability data. Does depletion of USP39 reduce cell viability or slow down the proliferation rate?
Response 2: We thank for the reviewer’s comment. We are sorry we did not mark accurately in Figure 1A. In fact, we performed CCK8 assays and calculated the proliferation rate relative to control group. The results showed that USP39 slowed down cell proliferation. We have revised the label of column chart.
-Material and methods section should be drastically improved.
Specifically:
-3 cloning strategy to generate pLVX-USP39 and pSIH-H1-shUSP9/PSMD14 is not reported.
Response 3: We have supplemented the detailed information for cloning strategy in the Material and Methods section.
4.-Virus infection is poorly described.
Response 4: We have supplemented the description for virus infection in the Material and Methods section.
5.- there is no description of the detection system for immunoblot analysis.
Rseponse 5: We have supplemented the description of detection system for immunoblot analysis in the Material and Methods section.
- There is no indication of buffer composition used in the silver staining and mass spectrometry sub-section (loading buffer 1x composition should be provided).
Response 6: Thanks for the comment. Actually, the loading buffer used in silver staining was purchased from CoWin Biosciences. We have supplemented the product item in the Material and Methods section.
7.-Primers used for qPCR assays should be included in the supplementary file.
Response 7: The primers information has been supplemented in the Supplementary Table 1.
8.- There is no information about how the authors performed GO and the KEGG analysis.
Response 8:We have complemented the description on GO and KEGG analysis in the Methods section.
- Line 214 – I will carefully rephrase this sentence because there is no demonstration that overexpression of USP39 can promote tumorigenesis of non-transformed cells.
Response 9: We appreciate the reviewer’s comment. We have revised the sentence in the manuscript (line 237).
- Line 226-228 – It is not clear if tissue samples analyzed for USP39 protein expression by IHC derive from patients treated with chemotherapy, radiation, both, or nothing. Otherwise, it is not possible to correlate more survival with chemotherapy resistance.
Response 10: This is an important point. IHC samples derived from ESCC patients across all stages of TNM. For advanced ESCC, multimodality therapies incorporating systemic chemotherapies and/or radiotherapy were used. Moreover, chemotherapy strategy is not consistent. So it is difficult to accurately analyze the correlation between USP39 expression and chemosensitivity. According to the reviewer’s comment, we deleted the description “especially in chemoresistance”.
11.-Figure 4E. The Flag panel should not be split. Moreover, I could not understand why there is a Flag signal in the IP control lane. Similarly, also USP39 pattern could be shown in the IP lanes.
Response 11:We appreciate the reviewer’s careful review. After checking the results of Western Blotting, we found that we pasted the wrong results in the Figure 4E. We have corrected it in Figure 4E. The USP39 is tagged by Flag, so USP39 pattern was represented by the Flag pattern.
12.- It is not clear why the authors focused on MCM3 only, the alternative splicing events that is regulated, and why there is a correlation between USP39 and MCM3 expression. Is the alternative exon encoded for a cryptic exon that leads to AS-NMD activation?
Response 12:We respectfully agree with this reviewer’s comment. Please see Response 2& 22. RIP-Seq showed that USP39 binds directly to the MCM2, MCM3, MCM4, MCM6, MCM7 and MCM9 etc mRNA, and RNA-Seq showed that USP39 regulates the splicing of MCM2, MCM3 and MCM10 etc. The results demonstrate that USP39 may be involved in the splicing of multiple members of MCM family, suggesting that it is not sufficient to select MCM3 for further investigation. Moreover, we only provide the preliminary evidence show that USP39 may regulate the alternate splicing. And no evidence to elucidate the underlying mechanism of USP39 regulating MCM3 splicing and prove that USP39 promotes ESCC malignancy by affecting the alternative splicing events of MCM3. According to all reviewers’ comments, we decided to remove MCM3 results.
13.- Differences in Fig. 5D and 5E are very slight. Did the author investigate the AS event also through PCR and gel electrophoresis to discriminate among AS variants?
Response 13:Thank you for your careful review. Currently,we have decided to remove the preliminary data on MCM3 from the manuscript.
Minor comments:
14.- Gene and transcript names should be in italics (i.e. line 38 and in the figures);
Response 14:Thank you for your careful review. We have checked the manuscript and made the corresponding revision.
15.- No error bars for PSMD14 condition are visible in Figure 1A.
Response 15: We appreciate the reviewer’s comment. Invisible error bars are due to the proliferation rate is too near among different repetitions for PSMD14 group.
16.- Cloning procedures in the material and methods section could be improved;
Response 16: We appreciate the reviewer’s comment. The cloning procedures have supplemented in the Material and Methods 2.1.
17.-Figure 3C, 4A. Protein size is missing.
Response 17:We thank for the reviewer’s comment. We have complemented the MW in all blot.
18.-To properly evaluate Flow cytometry data, the gating strategy used for the analysis should be included.
Response 18: We thank for the reviewer’s comment. The gating strategy used for the apoptosis analysis is showed in Response Figure 2.
Response Figure 2. The gating strategy for the apoptosis analysis. Please see the attached file.
19- I will include in the main text also the common name of DPP.
Response 19: We thank for the reviewer’s comment. We have included the common name of DDP in the manuscript.
20-Supplementary Figure 1E. The quality of the colony formation assay images is very poor. It is hard to see colonies in shUSP39 #1 and #2 as quantified in the plot on the right.
Response 20: We thank for the reviewer’s comment. It is hard to see since the colonies formed by KYSE30 cells are too small. We have showed more clear pictures.
21-Figures 4A and 4b are more appropriate as supplementary panels.
Response 21: We thank for the reviewer’s comment. We would like to emphasize that Figure 4A&4B showed that the efficiency of USP39 overexpression and USP39-enriched proteins compared with the vector control, we were prone to reserve the panels in Figure 4A&4B.
22- There is no information about the statistical test used in Supplementary Fig 1 and 2.
Response 22: We appreciate for the reviewer’s comment. We have included the statistical test information used in Supplementary Figure 1&2.
23- The numerosity (n) of each group should be reported in each analysis
Response 23: We appreciated for the comment. We have added the numerosity of each group in the manuscript.
24.-Fig. 5A – A pie chart will be more informative. Also, the definition of Tes, Tss, ExonAll should be included in the figure legend.
Response 24: We thank for the reviewer’s comment. Please see Response 18. CDS is coding sequence, Tes is transcription end site, Tss is transcription start site, 3’UTR is 3’-untranslated region, 5’UTR is 5’-untranslated region. We have included the definition of Tes, Tss, ExonAll in the figure legend.
25- Fig. 5D and E. Please correct “soliced mRNA/mRNA”.
Response 25: We have corrected it.
26- Line 280 – It is not clear which GEO expression dataset has been used.
Response 26: We appreciate the reviewer’s comment. This comment is similar to the reviewer 1#. We download the expression data from GEO library, and the dataset we used is from GSE161533 file. As explained in Response 2, 22 &35, we have decided to delete the results on MCM3 and the data analysis using GSE161533 dataset have been removed.
In light of these improvements, we are optimistic that all of the concerns have been addressed to your satisfaction, and that you and the reviewer will consider this work appropriate for Genes. Thanks again for all of your efforts in handling our manuscript.

Reviewer 3 Report
This is an interesting paper by Zhu and co-workers evaluating the role of Deubiquitinase USP39 in the development of ESCC Malignancy via MCM3 Splicing. Authors documented that deubiquitinase USP39 promoted esophageal cancer cell proliferation, migration and invasion; USP39 drived chemoresistance in esophageal cancer via inhibition of cell apoptosis; USP39 interacted with spliceosome components: EFTUD2, PRPF3, SART1, DDX23 258 and hnRNPU; and USP39 affected MCM3 splicing in ESCC.
The authors concluded that USP39 may serve as an oncogenic factor in ESCC through promotion of tumor proliferation in vivo and in vitro, increase of invasion and migration of cancer cells and inhibition of the cell apoptosis with the treatment of DDP. This is an interesting study, methodologically and clinically valuable.
The study was well performed and the data were thoroughly analyzed and interpreted. The authors extensively discussed the results.
The idea and results presented in this manuscript may attract attention of some groups of researchers, especially those searching for new anti-cancer modalities.
There are a few major issues that need to be addressed:
Comment 1. The title of the manuscript may be somehow misleading for readers as authors focused mostly on the role of USP39 in cancer progression but not MCM3. The authors hardly documented the role of MCM3 in ESCC as they showed only the correlation between USP39 and MCM3 expression in ESCC (Fig. 5). In my eyes presented data do not indicate direct effects of MCM3 on cancer progression and chemoresistance related to USP39 but rather implication of USP39 in these processes.
Comment 2. Does inhibitory effects of USP39 silencing affect also non-cancer cells or are these effects seem to be exclusive for ESCC. In other words, could USP39 be regarded as a specific molecular target in anticancer therapy. I would like to see effects of USP39 inhibition on proliferation and MCM3 expression in normal cells.
Comment 3. Colony formation assay technique used in the study (Fig. 2E and F) was not described in the Materials and Methods section.
Comment 4. Western blot results shown in Fig. 3C as a representative blot could be quantified, average of independent experiments calculated and presented in chart. It would be beneficial for readers to compare values of c-Caspase 3 and c-PARP in cancer and overexpressed-USP-39 cancer cells.
Comment 5. The authors persistently use abbreviations and terms without relevant explanation in the text. For example: although MCM3 is used in the title and many times throughout the text it is finally revealed at the end of the manuscript, in the Discussion section; study of ESCC chemoresistance with the use of DDP(?) and again no explanation – I guess cisplatin?
Comment 6. The microscopic images in Fig. 1E need scale bars or magnification description in the figure legends 1E.
Comment 7. Figure 1 legend does not match presented charts. Data described as
(C) Knockdown efficiency of UCHL5, USP48, VCPIP1 and PSMD14 in SW527 cells by qRT-PCR and E) Tumor volumes of UCHL5, USP48, VCPIP1 and PSMD14 silenced SW527 cells and control cells. n = 5 mice per group are not presented at all. On the other hand, ICH images and survival plots (Fig. 1 E and F, respectively) lack any description.
Minor issues:
Comment 1. Lines: 172. Mistyping error “Sirna” > siRNA.
Taken together, this paper by Zhu and co-workers represent a worthwhile contribution to the cancer research. It could be accepted with modifications as suggested.
Author Response
Response to Reviewer 3 Comments
Comments and Suggestions for Authors
This is an interesting paper by Zhu and co-workers evaluating the role of Deubiquitinase USP39 in the development of ESCC Malignancy via MCM3 Splicing. Authors documented that deubiquitinase USP39 promoted esophageal cancer cell proliferation, migration and invasion; USP39 drived chemoresistance in esophageal cancer via inhibition of cell apoptosis; USP39 interacted with spliceosome components: EFTUD2, PRPF3, SART1, DDX23 258 and hnRNPU; and USP39 affected MCM3 splicing in ESCC.
The authors concluded that USP39 may serve as an oncogenic factor in ESCC through promotion of tumor proliferation in vivo and in vitro, increase of invasion and migration of cancer cells and inhibition of the cell apoptosis with the treatment of DDP. This is an interesting study, methodologically and clinically valuable.
The study was well performed and the data were thoroughly analyzed and interpreted. The authors extensively discussed the results.
The idea and results presented in this manuscript may attract attention of some groups of researchers, especially those searching for new anti-cancer modalities.
There are a few major issues that need to be addressed:
Comment 1. The title of the manuscript may be somehow misleading for readers as authors focused mostly on the role of USP39 in cancer progression but not MCM3. The authors hardly documented the role of MCM3 in ESCC as they showed only the correlation between USP39 and MCM3 expression in ESCC (Fig. 5). In my eyes presented data do not indicate direct effects of MCM3 on cancer progression and chemoresistance related to USP39 but rather implication of USP39 in these processes.
Response 1: We do appreciate the reviewer’s comment. As Responded in 2, 22 and 35 above, currently, we only provide the preliminary evidence show that USP39 may regulate the alternate splicing. And no evidence to elucidate the underlying mechanism of USP39 regulating MCM3 splicing and prove that USP39 promotes ESCC malignancy by affecting the alternative splicing events of MCM3. So we removed the results on MCM3 in the current manuscript and made the corresponding revision.
Comment 2. Does inhibitory effects of USP39 silencing affect also non-cancer cells or are these effects seem to be exclusive for ESCC. In other words, could USP39 be regarded as a specific molecular target in anticancer therapy. I would like to see effects of USP39 inhibition on proliferation and MCM3 expression in normal cells.
Response 2: We thank for the reviewer’s comment. It is important to detect the effect on cell proliferation in non-cancer cells. Unfortunately, it is difficult to purchase culture medium and other growth factors for normal esophageal epithelial cells from abroad because of the COVID-19. We are so sorry that we can’t perform experiments to detect the effect on normal cell proliferation currently.
Comment 3. Colony formation assay technique used in the study (Fig. 2E and F) was not described in the Materials and Methods section.
Response 3: We appreciate the reviewer’s comment. We have included the colony formation assay in the Material and Methods 2.8
Comment 4. Western blot results shown in Fig. 3C as a representative blot could be quantified, average of independent experiments calculated and presented in chart. It would be beneficial for readers to compare values of c-Caspase 3 and c-PARP in cancer and overexpressed-USP-39 cancer cells.
Response 4: We thank for the reviewer’s comment. We have quantified the results of Western blot from the independent experiments and presented in chart in Fig. 3C.
Comment 5. The authors persistently use abbreviations and terms without relevant explanation in the text. For example: although MCM3 is used in the title and many times throughout the text it is finally revealed at the end of the manuscript, in the Discussion section; study of ESCC chemoresistance with the use of DDP(?) and again no explanation – I guess cisplatin?
Response 5: We appreciate the reviewer’s comment. We are sorry about the abbreviations without relevant explanation. The full name of MCM3 is Minichromosome Maintenance Complex Component 3 and the DDP is the abbreviation of cisplatin. We have checked the and included the full name of the abbreviations in the manuscript.
Comment 6. The microscopic images in Fig. 1E need scale bars or magnification description in the figure legends 1E.
Response 6: We thank for the reviewer’s comment. We have complemented the scale bars and magnification description of Fig. 1E.
Comment 7. Figure 1 legend does not match presented charts. Data described as
Response 7: We thank for the reviewer’s comment. We have modified the description of Figure 1 legend.
Comment 8. (C) Knockdown efficiency of UCHL5, USP48, VCPIP1 and PSMD14 in SW527 cells by qRT-PCR and E) Tumor volumes of UCHL5, USP48, VCPIP1 and PSMD14 silenced SW527 cells and control cells. n = 5 mice per group are not presented at all. On the other hand, ICH images and survival plots (Fig. 1 E and F, respectively) lack any description.
Response 8: We appreciate the reviewer’s comment. We have made the corresponding revision.
Minor issues:
Comment 1. Lines: 172. Mistyping error “Sirna” > siRNA.
Response: We thank for the reviewer’s comment. We have corrected it.
Taken together, this paper by Zhu and co-workers represent a worthwhile contribution to the cancer research. It could be accepted with modifications as suggested.
Response: We do appreciate this reviewer’s positive comment.
In light of these improvements, we are optimistic that all of the concerns have been addressed to your satisfaction, and that you and the reviewer will consider this work appropriate for Genes. Thanks again for all of your efforts in handling our manuscript.

Round 2
Reviewer 2 Report
1 – Figure 1 legend. I will suggest revising the use of the term transfect referred to a gene in the figure legend and within the text (line 203).
Furthermore, there is no information on the test used for the Kaplan-Meier analysis.
2 – I agree that indicating the relative proliferation rate is more accurate. However, I will include how this ratio is calculated in the method section.
3, 4, 5, 6, 7, 8 – I recognize the improvement of the method section.
9 – Ok.
10 – Ok.
11 – The quality of Fig. 4e has been improved. But, I do not understand the reason to have the Flag blotting on two different panels.
14- Minor comment 14 has not been addressed ( see again line 40).
15 – I disagree with the author's reply. Even if data are infinitely close to each other, the error bar should be visible and lay on the top of the column.I will also include a comment on the lack of PARP expression in cells overexpressing USP39, which is partially restored by DDP treatment.
16 – Ok.
17 – Ok.
18 – This data could be included as Supplementary Information to increase their accessibility to the readers.
19 – Ok.
20 – Ok.
21- I agree; it was only a suggestion.
22 – Ok.
23 - Ok.
24 – Ok.
25 – Ok.
I will suggest carefully revising the grammar language throughout the manuscript.
i.e. lines 13, 16, 24, 339.
In the revision version of the manuscript, the authors properly responded to most of the comments and suggestions raised in the first round. I agree with the author's decision to remove the MCM3 data from the manuscript.
However, in my opinion, the authors do not properly investigate the contribution of USP39 to alternative splicing regulation in ESCC (Result section 3.5).
The claims in this last section should be better supported. In particular: the authors focus their attention on the RIP-seq analysis without properly analyzing the RNA-seq data that are more relevant to claim the role of USP39 in the regulation of AS.
There is no information on transcripts differentially spliced or deregulated gene expression levels (except lane 334 in the discussion section). Specifically:
-Which are the gene expression levels mostly perturbed upon USP39 overexpression?
- Which are the AS events mostly deregulated upon USP39 overexpression?
- Which are Gene Ontology categories enriched in genes in which USP39 AS events are present (it could be different from categories found through Rip-seq analysis)?
- It will be useful to validate some targets in overexpressing and USP39-silenced cells as it has been done for the mass spec analysis of Flag-USP39 pull-down.
This is important because the authors claim that: "Mechaniscally, we provide evidence for a role for USP39 in alternative splicing regulation." (abstract line 16).
Furthermore, the authors claim to have identified 3 candidate USP39-spliced transcripts from a combined analysis of RIP-seq and RNA-seq data. However, the intercept in figure 5C show 4 candidates.
My suggestion is to drastically improve the characterization of the USP39 role in alternative splicing by adequately analyzing and reporting the enormous amount of data generated by the authors.
Author Response
Response to Reviewer 2 Comments
1 – Figure 1 legend. I will suggest revising the use of the term transfect referred to a gene in the figure legend and within the text (line 203). Furthermore, there is no information on the test used for the Kaplan-Meier analysis.
Response 1: We thank for the reviewer’s comment. We have made the corresponding revision. We performed Kaplan-Meier analysis by log rank test and we have supplemented it in Figure 1 legend.
2 – I agree that indicating the relative proliferation rate is more accurate. However, I will include how this ratio is calculated in the method section.
Response 2: The average absorbance at the control group was designated as 1. The relative cell proliferation rate is the ratio of the absorbance of each experimental group to the average of the control group. We have included the description in the method section.
3, 4, 5, 6, 7, 8 – I recognize the improvement of the method section.
9 – Ok.
10 – Ok.
11 – The quality of Fig. 4e has been improved. But, I do not understand the reason to have the Flag blotting on two different panels.
Response 11: We respectfully agree with the reviewer’s comment. Since the enrichment efficiency of Flag antibody is very high, the disparity of the signal intensity of Flag blotting between Input and IP group is very big. We therefore cut the membrance into two parts and detect the signal at the different conditions. We are sorry for not providing the Flag blotting on the same panel.
14- Minor comment 14 has not been addressed ( see again line 40).
Response 14: Thank you for your careful review. We have checked the manuscript and made the corresponding revision.
15 – I disagree with the author's reply. Even if data are infinitely close to each other, the error bar should be visible and lay on the top of the column. I will also include a comment on the lack of PARP expression in cells overexpressing USP39, which is partially restored by DDP treatment.
Response 15: Thank for the reviewer’s comment. Actually, the SD of USP9Y and PSMD14 is 0.000, so error bar is invisible.
We do appreciate the reviewer’s comment on the PARP expression. We are confused for the lack of PARP expression in cells overexpressing USP39 and we can’t provide the explanation for the PARP expression changes in cells overexpressing USP39. We therefore remove the panels at 0h in Figure 3C.
16 – Ok.
17 – Ok.
18 – This data could be included as Supplementary Information to increase their accessibility to the readers.
Response 18: We do appreciate for the reviewer’s comment. We have included the results in Supplementary Figure 3.
19 – Ok.
20 – Ok.
21- I agree; it was only a suggestion.
22 – Ok.
23 - Ok.
24 – Ok.
25 – Ok.
I will suggest carefully revising the grammar language throughout the manuscript. i.e. lines 13, 16, 24, 339.
Response: Thanks for the reviewer’s comment. We have carefully revised the grammar language in the manuscript.
In the revision version of the manuscript, the authors properly responded to most of the comments and suggestions raised in the first round. I agree with the author's decision to remove the MCM3 data from the manuscript. However, in my opinion, the authors do not properly investigate the contribution of USP39 to alternative splicing regulation in ESCC (Result section 3.5). The claims in this last section should be better supported. In particular: the authors focus their attention on the RIP-seq analysis without properly analyzing the RNA-seq data that are more relevant to claim the role of USP39 in the regulation of AS. There is no information on transcripts differentially spliced or deregulated gene expression levels (except lane 334 in the discussion section). Specifically:
-Which are the gene expression levels mostly perturbed upon USP39 overexpression?
Response: Actually, we performed RNA-Seq in USP39-silenced cells and control cells. Differential expression analysis showed that 300 genes were up-regulated, while 288 genes were down-regulated upon USP39 silencing. Among them, YTHDF2, PTPN11, TSEN5 and HAS3 etc. are obviously dow-regulated. And YTHDF2 is a m6A reader protein, PTPN11 encoded the protein tyrosine phosphatas SHP2, TSEN5 is a tRNA splicing endonuclease, and HAS3 is a member of the hyaluronan synthase family of enzymes, which gives rise to different transcripts through alternative splicing.
- Which are the AS events mostly deregulated upon USP39 overexpression?
Response: Combined with AS events analysis and experimental validation, we confirmed USP39 regulated the alternative splicing of ESPR2, FOXK2, CCT7, HNRNPC and NCOA4. We included the results in update Figure 5D.
- Which are Gene Ontology categories enriched in genes in which USP39 AS events are present (it could be different from categories found through Rip-seq analysis)?
Response: Thanks for your concern. We have performed GO analysis and included the results in Figure 5C.
- It will be useful to validate some targets in overexpressing and USP39-silenced cells as it has been done for the mass spec analysis of Flag-USP39 pull-down.
Response: This is an important point. Combined with AS events analysis and experimental validation, we confirmed USP39 regulated the alternative splicing of ESPR2, FOXK2, CCT7, HNRNPC and NCOA4 . We included the results in update Figure 5D.
This is important because the authors claim that: "Mechaniscally, we provide evidence for a role for USP39 in alternative splicing regulation." (abstract line 16). Furthermore, the authors claim to have identified 3 candidate USP39-spliced transcripts from a combined analysis of RIP-seq and RNA-seq data. However, the intercept in figure 5C show 4 candidates. My suggestion is to drastically improve the characterization of the USP39 role in alternative splicing by adequately analyzing and reporting the enormous amount of data generated by the authors.
Response: We realized that the evidence is not sufficient to supporting the role for USP39 in alternative splicing regualtion. According to the reviewer’s suggestion, we complemented GO analysis results of AS events and validated AS events by RT-PCR in Figure 5. Meanwhile, we removed the differential expression gene in the original Figure 5C. And we identified 3981 binding peaks by RIP-Seq and 3496 AS events by RNA-Seq. The integrated analysis of RIP-Seq and AS events showed that USP39 regulate hundreds of AS events by direct RNA binding. Accordingly, we have modified the description in the Results and Discussion section.
Reviewer 3 Report
Revised paper by Zhu and co-workers "The Deubiquitinase USP39 Promotes ESCC Malignancy as a splicing factor" represent a worthwhile contribution to the cancer research. I recommend the manuscript for further publication process.
Author Response
Thanks.
Round 3
Reviewer 2 Report
The authors have fully addressed my concerns. Even without including some of the initial data, I think the work has acquired more robustness. Overall, it will contribute to a better knowledge of the role of USP39, and in general alternative splicing, in esophageal squamous cell carcinomas.